# Is the Morphological Subtype of Extra-Pulmonary Neuroendocrine Carcinoma Clinically Relevant?

**DOI:** 10.3390/cancers13164152

**Published:** 2021-08-18

**Authors:** Melissa Frizziero, Alice Durand, Rodrigo G. Taboada, Elisa Zaninotto, Claudio Luchini, Bipasha Chakrabarty, Valérie Hervieu, Laura C. L. Claro, Cong Zhou, Sara Cingarlini, Michele Milella, Thomas Walter, Rachel S. Riechelmann, Angela Lamarca, Richard A. Hubner, Wasat Mansoor, Juan W. Valle, Mairéad G. McNamara

**Affiliations:** 1Division of Cancer Sciences, Faculty of Biology Medicine and Health, University of Manchester, Manchester M13 9PL, UK; melissa.frizziero@cruk.manchester.ac.uk (M.F.); angela.lamarca@nhs.net (A.L.); richard.hubner@nhs.net (R.A.H.); juan.valle@nhs.net (J.W.V.); 2Department of Medical Oncology, The Christie NHS Foundation Trust, Manchester M20 4BX, UK; was.mansoor@nhs.net; 3Department of Gastroenterology and Medical Oncology, Edouard Herriot Hospital, Hospices Civils de Lyon, 69003 Lyon, France; alice.durand@chu-lyon.fr (A.D.); thomas.walter@chu-lyon.fr (T.W.); 4Department of Clinical Oncology, A. C. Camargo Cancer Center, São Paulo 01509-010, Brazil; gomes.taboada@gmail.com (R.G.T.); rachel.riechelmann@accamargo.org.br (R.S.R.); 5Department of Medical Oncology, University Hospital of Verona, 37134 Verona, Italy; elisa.zaninotto@gmail.com (E.Z.); cingarlini@icloud.com (S.C.); michele.milella@univr.it (M.M.); 6Department of Diagnostics and Public Health, Section of Pathology, University and Hospital Trust of Verona, 37134 Verona, Italy; claudio.luchini@univr.it; 7Department of Pathology, The Christie NHS Foundation Trust, Manchester M20 4BX, UK; bipasha.chakrabarty@nhs.net; 8Department of Pathology, Edouard Herriot Hospital, Hospices Civils de Lyon, 69003 Lyon, France; valerie.hervieu@chu-lyon.fr; 9Department of Pathology, A. C. Camargo Cancer Center, São Paulo 01509-010, Brazil; laura.claro@accamargo.org.br; 10Cancer Biomarker Centre, Cancer Research UK Manchester Institute, University of Manchester, Alderley Park SK10 4TG, UK; cong.zhou@cruk.manchester.ac.uk

**Keywords:** extra-pulmonary neuroendocrine carcinoma, small cell, non-small cell, morphology

## Abstract

**Simple Summary:**

Neuroendocrine carcinomas (NECs) represent the most aggressive subgroup of neuroendocrine neoplasms. Around 90% of NECs arise from the lung. The minority of NECs originating outside of the lung are called extra-pulmonary (EP)-NECs. Most patients with EP-NECs are diagnosed at an advanced stage (incurable) and have a life expectancy of months; platinum-based chemotherapy or best supportive care are the only options for these patients. However, response to platinum-based chemotherapy and prognosis vary largely within this patient population. Previous studies have shown that such variability depends on the site of origin and the Ki-67 index (which is an indicator of how quickly cancer cells proliferate). The present study found that the morphological subtype—small cell (SC) or non-small cell (non-SC)—is another contributing factor. In fact, patients with an advanced-stage non-SC EP-NEC respond less to platinum-based chemotherapy and have shorter survival than patients with an advanced-stage SC EP-NEC. Alternative treatments should be considered for this subgroup.

**Abstract:**

Extra-pulmonary neuroendocrine carcinomas (EP-NECs) are lethal cancers with limited treatment options. Identification of contributing factors to the observed heterogeneity of clinical outcomes within the EP-NEC family is warranted, to enable identification of effective treatments. A multicentre retrospective study investigated potential differences in “real-world” treatment/survival outcomes between small-cell (SC) versus (vs.) non-SC EP-NECs. One-hundred and seventy patients were included: 77 (45.3%) had SC EP-NECs and 93 (54.7%) had non-SC EP-NECs. Compared to the SC subgroup, the non-SC subgroup had the following features: (1) a lower mean Ki-67 index (69.3% vs. 78.7%; *p* = 0.002); (2) a lower proportion of cases with a Ki-67 index of ≥55% (73.9% vs. 88.7%; *p* = 0.025); (3) reduced sensitivity to first-line platinum/etoposide (objective response rate: 31.6% vs. 55.1%, *p* = 0.015; and disease control rate; 59.7% vs. 79.6%, *p* = 0.027); (4) worse progression-free survival (PFS) (adjusted-HR = 1.615, *p* = 0.016) and overall survival (OS) (adjusted-HR = 1.640, *p* = 0.015) in the advanced setting. Within the advanced EP-NEC cohort, subgroups according to morphological subtype and Ki-67 index (<55% vs. ≥55%) had significantly different PFS (adjusted-*p* = 0.021) and OS (adjusted-*p* = 0.051), with the non-SC subgroup with a Ki-67 index of <55% and non-SC subgroup with a Ki-67 index of ≥55% showing the best and worst outcomes, respectively. To conclude, the morphological subtype of EP-NEC provides complementary information to the Ki-67 index and may aid identification of patients who could benefit from alternative first-line treatment strategies to platinum/etoposide.

## 1. Introduction

Extra-pulmonary neuroendocrine carcinomas (EP-NECs) are aggressive epithelial cancers with immunohistochemical expression of neuroendocrine (NE) markers (chromogranin A, synaptophysin or neuron cell adhesion molecule), and a proliferation (Ki-67) index of >20% [1]. In addition, EP-NECs lack the typical organoid-like growth pattern of low-grade, well-differentiated neuroendocrine tumours (WD-NETs) and are therefore defined as poorly differentiated. Similar to their pulmonary counterparts, EP-NECs can exhibit a “small-cell” (SC) morphology; diffuse sheets of cells with scant cytoplasm and fusiform nuclei with inconspicuous nucleoli and finely granular chromatin or a “large-cell” morphology; nests- or trabeculae-like patterns of round/polygonal cells, with moderate amounts of cytoplasm and large nuclei with prominent nucleoli and vesicular chromatin [1].

EP-NECs are rare diseases, with an annual incidence of ~1/100,000 individuals in the United States according to the Surveillance, Epidemiology, and End Results (SEER)-18 registry (1973–2012) [2]. However, their incidence has been steadily rising [3], mainly as a result of the refinement of diagnostic methods and the increased awareness of this diagnosis within the scientific community.

Patients with an EP-NEC diagnosis mostly have metastatic disease at presentation and have an average life expectancy of less than 12 months [2]. Treatment options for these patients are limited; surgery remains the mainstay of treatment in the localised setting; platinum-based chemotherapy is the only standard-of-care first-line palliative treatment and has not changed for the past three decades [4]. Although radiological responses are observed in up to ~70% of patients receiving first-line platinum-based chemotherapy, disease progression occurs rapidly within months of the start of the treatment (median progression-free survival (PFS): 4–9 months) [5,6]. Several chemotherapy regimens have been investigated after the failure of platinum-based chemotherapy in small retrospective studies or non-randomised trials [7], but none have become standard practice so far.

The identification of effective treatment strategies for EP-NECs has been hampered by their low incidence, which makes the conduct of large randomised clinical trials challenging, and the paucity of knowledge of their molecular drivers (besides frequent *TP53* and *RB1* loss) [8]. Furthermore, whilst EP-NECs have historically been approached clinically as a single disease, evidence from large datasets points towards wide variability in survival and treatment outcomes within the EP-NEC family [2,9], which is suggestive of underlying biological heterogeneity. The anatomical site of origin and a Ki-67 threshold of 55% have emerged as key contributing factors to such variability. In the SEER-18 cohort [2] including 14,732 patients with NEC from any EP anatomical site (any disease stage) and the NORDIC study [9] including 305 patients with advanced-stage NEC from the gastro-entero-pancreatic (GEP) tract or of unknown origin (cancer of unknown primary, CUP), the site of origin was an independent prognosticator of overall survival (OS) (median OS ranged from 2.5 months for CUP-NECs and liver-NECs to 25 months for small bowel-NECs). In addition, in the NORDIC study [9], a Ki-67 index of <55% was associated with a significantly longer OS, but a lower likelihood of response to platinum-based chemotherapy compared to patients with a Ki-67 index of ≥55%. The prognostic significance of the Ki-67 index (<55% versus (vs.) ≥55%) was corroborated in subsequent studies [7,10,11,12]. In pulmonary NECs, the morphological subtype identifies two clinically distinct entities with only partially overlapping molecular landscapes, i.e., small-cell lung cancer (SCLC) and large-cell pulmonary neuroendocrine carcinoma (LCPNEC) [13]. Whether the morphological subtype is a source of clinical and biological heterogeneity within the EP-NEC family also remains a matter of debate. In the SEER-18 cohort, patients with SC EP-NECs (any disease stage) had a significantly shorter OS than patients with non-SC EP-NECs (any disease stage) [2]. However, other studies in EP-NECs reported no prognostic significance of the morphological subtype [9,14,15].

The present study investigates differences in treatment and survival outcomes between SC and non-SC subgroups, after expert pathological review, in one of the largest retrospective EP-NEC cohorts in the published literature. The aim was to address the question as to whether the morphological subtype is relevant for patient management and therefore should this be incorporated in the diagnostic work-up of patients with EP-NECs as standard practice.

## 2. Materials and Methods

Consecutive patients with an EP-NEC diagnosis and available formalin-fixed paraffin-embedded (FFPE) tumour tissue were identified retrospectively through medical records at four specialised centres for neuroendocrine neoplasms (NENs), i.e., The Christie NHS Foundation Trust (Manchester, UK), Edouard Herriot Hospital, Hospices Civils de Lyon (Lyon, France), A. C. Camargo Cancer Center (São Paulo, Brazil) and Policlinico G.B. Rossi, University Hospital of Verona (Verona, Italy). The study received local ethics or audit committee approval by each participant institution. Informed signed consent from individual patients was not required. Tumour tissue was reviewed by pathologists with expertise in NENs (C.L., B.C., V.H. and L.C.L.C.), and only cases meeting the 2019 World Health Organisation (WHO) diagnostic criteria for EP-NEC [1] were included. In line with these criteria, all tumours included must have shown the immunohistochemical expression of at least one of the following NE markers, i.e., chromogranin A, synaptophysin and neural cell adhesion molecule (NCAM)/CD56, in at least 70% of the tumour mass. Mixed neuroendocrine non-neuroendocrine neoplasms (MiNENs) or grade-3 (G3)-WD-NETs, as per 2019 WHO classification [1], were excluded. The morphological subtype was classified as SC or non-SC (which included large-cell morphology and also those with intermediate features between the SC and the large cell), based on the morphological criteria applied for lung NECs [16]. CUP-NECs were included, providing a primary origin from the lung could be ruled out, based on radiological investigations and immunohistochemical profile. Classification of equivocal cases was discussed among pathologists via videoconference; when an agreement was not reached, the case was excluded (e.g., uncertainty in relation to classifying a G3 NEN as an NEC or G3-WD-NET, or an EP-NEC as an SC or non-SC). Disease stage at diagnosis was reported in compliance with the most recent (8th) edition of the American Joint Committee on Cancer Tumour, Nodes, Metastasis (TNM) classification for adenocarcinomas from the same sites of origin [17]. Patients were further classified as belonging to the “potentially curable” or “advanced: subgroup, based on whether they were treated with curative intent or not”. The “advanced” subgroup also included those patients who developed disease progression after initial curative treatment. Best overall response to chemotherapy, radiotherapy, chemo-radiotherapy or immunotherapy was defined as the best response recorded from the start until the end of a line of treatment in accordance with the Response Evaluation Criteria in Solid Tumours (RECIST) v1.1 [18]. Physicians with oncology training and experience in RECIST calculation (by M.F., E.Z., A.D. and R.G.T.) reviewed imaging reports and extracted measurements of tumour lesions to classify tumour responses according to RECIST v1.1 [18]. The support of a specialist radiologist was sought for the interpretation of equivocal lesions, where appropriate. When patients did not undergo any radiological assessment after the start of a line of treatment due to physical deterioration/increase in symptom burden, best response was reported as progressive disease (PD) based on clinical judgment. Stable disease (SD) was defined as neither sufficient shrinkage to qualify for partial response nor sufficient increase to qualify for PD, taking as a reference the smallest sum diameters while on study, as per Eisenhauer et al. [18]. Disease-free survival (DFS) for the “potentially curable” subgroup was estimated from the date of the initial diagnosis to the date of the radiological evidence of tumour recurrence, death or last follow-up. Progression-free survival (PFS) for the “advanced” subgroup was estimated from the date of the start of the active palliative treatment to the date of the radiological and/or unequivocal clinical evidence of disease progression (if radiological tests were not performed), death or last follow-up. Patients who were not eligible for active palliative treatment were not included in the PFS analysis. OS was calculated from the date of the initial diagnosis or the date of the disease recurrence to the date of death or last follow-up. Associations between the morphological subtype and other clinico-pathological characteristics were interrogated by applying the Pearson’s chi-squared or Fisher’s exact test for categorical variables, and the independent two-sample *t*-test or Mann–Whitney U test for continuous variables (the assumption of normality for continuous variables was tested by graphical representation, skewness analysis and evaluation of standard deviation). Kaplan–Meier analysis was employed, and differences in survival (DFS, PFS and OS) were evaluated using log-rank tests. Median survival and median follow-up time were summarised. Univariable (for DFS, PFS and OS) and multivariable analyses (for PFS and OS) were conducted using Cox-regression models to interrogate the prognostic value of clinico-pathological characteristics. Interactions between significantly associated clinico-pathological characteristics were included in the models. These analyses were carried out following REMARK guideline [19]. Descriptive and inferential statistics were performed using GraphPad Prism v8.4.2. and IBM SPSS Statistics 25.

## 3. Results

Archival tumour tissues from 170 patients (identified over a 25-year time period, from April 1994 to March 2019) were confirmed to be in keeping with an EP-NEC diagnosis as per 2019 WHO classification criteria [1] and could be classified as SC (number of patients (*n*) = 77; 45.3%) or non-SC (*n* = 93; 54.7%) (Figure 1). Clinico-pathological characteristics for the whole population and according to the morphological subtype are presented in Table 1. Notably, the Ki-67 index was significantly lower for the non-SC subgroup compared to for the SC subgroup (*p* = 0.002). In addition, NECs of oesophageal or oesophago-gastric origin were more commonly of SC morphology (*p* = 0.001).

### 3.1. “Potentially Curable” Subgroup

Patients with “potentially curable” EP-NECs (*n* = 39) were predominantly treated with surgery in combination with adjuvant and/or neoadjuvant chemotherapy (*n* = 19; 48.7%). When chemotherapy was administered with curative intent in combination with surgery or radiotherapy (*n* = 27), platinum/etoposide was the most commonly chosen regimen (18/27; 66.7%). Treatment choices did not significantly differ between patients with SC and non-SC EP-NECs (Appendix A).

After a median follow-up time of 56.61 months, 30 (76.9%) patients relapsed, and 23 (59.0%) died. The median DFS and OS for this subgroup were 12.10 months (95% confidence interval (95% CI): 7.28–16.92) and 32.51 months (95% CI: 11.78–53.24), respectively. On univariable analysis, there was no statistically significant difference in either survival outcomes between the SC (*n* = 13; 33.3%) and non-SC (*n* = 26; 66.7%) subgroups, with median DFS values of 10.12 and 12.10 months, respectively (hazard ratio (HR) = 1.46 (95% CI: 0.71–3.02), *p* = 0.309) and median OS values of 31.56 and 32.51 months, respectively (HR = 1.64 (95% CI: 0.69–3.87), *p* = 0.262).

### 3.2. “Advanced” Subgroup

The “advanced” subgroup (*n* = 161) included 131 patients with incurable disease at diagnosis and 30 patients from the “potentially curable” subgroup who relapsed after initial curative treatment. Information on treatment was available for 159 patients (98.8%) and is summarised in Appendix A and Appendix A. The most commonly offered treatments were chemotherapy with platinum/etoposide in the first-line setting (*n* = 118/159; 74.2%) and a fluoropyrimidine/irinotecan combination in the second-line setting (*n* = 47/124; 37.9%). There was no significant difference in the proportion of patients treated with either regimen between the two subgroups per morphological subtype: first-line platinum/etoposide (SC = 58/73 (79.5%) vs. non-SC = 60/86 (69.8%); *p* = 0.164) and second-line fluoropyrimidine/irinotecan (SC = 20/58 (34.5%) vs. non-SC = 27/66 (40.9%); *p* = 0.462).

#### 3.2.1. Treatment Response in the “Advanced” Subgroup

Information on the best response to first-line chemotherapy alone or in combination with radiotherapy was available for 131 patients (60 with SC and 71 with non-SC EP-NEC). Complete response (CR) was reported in 8 (13.3%) patients with an SC EP-NECs and 5 (7.0%) with non-SC EP-NECs, partial response (PR) in 22 (36.7%) patients with SC EP-NECs and 20 (28.2%) patients with non-SC EP-NECs, SD in 17 (28.3%) patients with SC EP-NECs and 18 (25.4%) with non-SC EP-NECs, PD in 13 (21.7%) patients with SC EP-NECs and 28 (39.4%) with non-SC EP-NECs. There was no significant difference in objective response rate (ORR) (CR + PR) between the SC and non-SC subgroups (50.0% vs. 35.2%; *p* = 0.088). However, the disease control rate (DCR) (CR + PR + SD) was significantly higher for the SC subgroup compared to for the non-SC subgroup; (78.3% vs. 60.6%; *p* = 0.029). Within this sub-population, there was no significant difference in ORR (*p* = 0.210) or DCR (*p* = 0.203) according to the Ki-67 index (<55% vs. ≥55%) (Figure 2).

The best response to first-line platinum/etoposide alone or in combination with radiotherapy was reported for 106 patients; 49 with SC EP-NECs and 57 with non-SC EP-NECs. A similar proportion of patients received radiotherapy in the two subgroups; 4.1% for SC and 7.0% for the non-SC subgroup (*p* = 0.684). CR occurred in 6 (12.2%) patients with SC EP-NECs and 4 (7.0%) with non-SC EP-NECs, PR in 21 (42.9%) patients with SC EP-NECs and 14 (24.6%) with non-SC EP-NECs, SD in 12 (24.5%) patients with SC EP-NECs and 16 (28.1%) with non-SC EP-NECs, and PD in 10 (20.4%) patients with SC EP-NECs and 23 (40.4%) with non-SC EP-NECs. Both the ORR and the DCR were significantly higher in the SC subgroup compared to in the non-SC subgroup (ORR: 55.1% vs. 31.6% (*p* = 0.015); and DCR: 79.6% vs. 59.7% (*p* = 0.027); Figure 2). Within this subpopulation, there was no difference in ORR (*p* = 0.392) or DCR (*p* = 0.322) according to the Ki-67 index (<55% vs. ≥55%).

In patients treated with a fluoropyrimidine/irinotecan combination in the first or second-line setting (*n =* 48), both the ORR and the DCR were higher for the non-SC subgroup (*n =* 27) compared to in the SC subgroup (*n =* 21), but the difference was not statistically significant (ORR: 22.2% vs. 9.5% (*p* = 0.437); and DCR: 40.7% vs. 33.3% (*p* = 0.599; Figure 2). Within this subpopulation, neither ORR (*p* = 0.568) nor DCR (*p* = 0.073) differed according to the Ki-67 index (<55% vs. ≥55%).

#### 3.2.2. Survival Outcomes in the “Advanced” Subgroup

In the “advanced” subgroup, 146 patients were evaluated for PFS and 160 patients were assessed for OS. After a median follow-up time of 47.57 months, 130/146 (89.0%) patients had disease progression, and 128/160 (80.0%) had died. The overall median PFS and OS were 5.72 months (95% CI: 4.83–6.61) and 12.52 months (95% CI: 9.64–15.40), respectively. There was no statistically significant difference in PFS or OS according to the morphological subtype on univariable analysis with a median PFS of 6.21 months for the SC subgroup (*n* = 66; 45.2%) and a median PFS of 4.93 months for the non-SC subgroup (*n* = 80; 54.8%) (HR=0.79 (95% CI: 0.56–1.12); *p* = 0.180). OS was numerically longer for the SC subgroup compared to for the non-SC subgroup, with a median OS of 16.04 months for the SC subgroup (*n* = 74; 46.3%) and a median OS of 10.52 months for the non-SC subgroup (*n* = 86; 53.7%) (HR = 0.74 (95% CI: 0.52–1.05); *p* = 0.095; Figure 3, Table 2). Two significant prognostic factors for PFS—The Eastern Cooperative Oncology Group performance status (ECOG PS) (≥2 vs. 0–1; HR = 2.62, *p* < 0.005) and the Ki-67 index (≥55% vs. <55%; HR = 1.67, *p* = 0.040)—and one significant prognostic factor for OS; ECOG PS (≥2 vs. 0–1; HR = 2.83, *p* < 0.005) were identified by univariable analysis (Table 2). In the multivariable analysis, the impact of the morphological subtype on survival outcomes was adjusted for the ECOG PS and the Ki-67 index. The non-SC subtype was an independent prognostic factor for both worse PFS (adjusted-HR = 1.62; *p* = 0.016) and OS (adjusted-HR = 1.64; *p* = 0.015), whereas a Ki-67 index of ≥55% (significantly enriched in the SC subgroup; Table 1) was an independent prognostic factor for worse PFS (adjusted-HR = 1.80; *p* = 0.028) and showed a trend towards worse OS (adjusted-HR = 1.63; *p* = 0.067). Due to the significant association between the morphological subtype and the Ki-67 index (<55% vs. ≥55%) (Table 1), their interaction was interrogated and was significant for neither PFS nor OS (Table 2). As these two factors impacted independently on survival outcomes of patients with advanced EP-NEC, they were jointly applied to stratify this patient cohort into the following subgroups: (A) non-SC EP-NECs with a Ki-67 index of <55% (*n =* 21); (B) non-SC EP-NECs with a Ki-67 index of ≥55% (*n =* 60); (C) SC EP-NECs with a Ki-67 index of <55% (*n =* 5); (D) SC EP-NECs with a Ki-67 index of ≥55% (*n =* 55) (Figure 4). This stratification resulted statistically significant for PFS (log-rank *p* = 0.007) and “borderline”-significant for OS (log-rank *p* = 0.054) even when after adjustment for ECOG PS (adjusted-*p* = 0.021 for PFS and 0.051 for OS). In addition, it was observed that subgroups A and B had the best and worst outcomes, respectively.

Survival outcomes according to the morphological subtype, as well as univariable and multivariable analyses in the sub-population of patients who received at least one line of palliative chemotherapy alone or in combination with radiotherapy (*n* = 136 evaluated for PFS; *n* = 143 evaluated for OS), and in the sub-population of patients who received platinum/etoposide alone or in combination with radiotherapy in the first-line setting (*n* = 109 evaluated for PFS; *n* = 115 evaluated for OS), are presented in Appendix A.

## 4. Discussion

Patients with EP-NEC are currently treated following the treatment paradigm for SCLC, based on the assumption of a biological similarity with their pulmonary counterpart. However, their clinical and epidemiological behaviours are apparently distinct; a multicentre retrospective study with central pathology review reported a higher response rate to first-line platinum-based chemotherapy (86.8% vs. 44.6%; *p* < 0.001) and higher tobacco history (98.8% vs. 46.7%; *p* < 0.001) for patients with advanced SCLC compared to for those with EP-NEC [20]. Additionally, there is still limited knowledge of the EP-NEC biology [6]. A number of studies have attempted to elucidate the molecular landscape of EP-NECs over the past few years [21,22,23,24,25], as these cancers have been gaining more attention in the scientific community. Accumulating evidence indicates that besides a common element of NE biology (e.g., high prevalence of *TP53* and *RB1* loss), EP-NECs exhibit typical molecular traits of adenocarcinomas from the same sites of origin [8]. This raises the question as to whether treatment strategies which are established for the latter could also find application in EP-NECs. In the CIRCAN-NEC study [24], reporting on circulating tumour DNA (ctDNA) analysis in patients with advanced-stage GEP- or CUP-NEC, the most commonly mutated genes were *TP53* and *RB1*, but also genes associated with GEP adenocarcinoma pathogenesis (e.g., *KRAS*, *BRAF* and *APC*). Interestingly, those cases classified as “adenocarcinoma-like” based on their ctDNA profile showed less durable responses to the first-line platinum/etoposide chemotherapy, compared to cases which did not harbour any adenocarcinoma-associated mutations.

The EP-NEC family is known to be characterised by wide inter-patient variability with regard to prognosis and response to treatment. Therefore, there is a need for biomarkers that can guide patient stratification and aid in the identification of tailored treatments. Lamarca et al. [26] developed a combined prognostic score for patients with GEP-NEC including five variables, i.e., presence of liver metastases, alkaline phosphatase (ALP), lactate dehydrogenase (LDH), ECOG PS and Ki-67 (≤80% vs. >80%). The so-called GI-NEC score was initially derived in a training cohort of 109 patients, where it discriminated between patients with significantly different OS with high performance, and was subsequently validated in an external (*n =* 184) and a prospective cohort (*n =* 20). This study posits that the GI-NEC score could be applied in routine practice to predict whether the clinical course of a patient with a newly diagnosed GEP-NEC will be more or less favourable, enabling the delivery of more personalised management, and it could also be integrated in the design of clinical trials in GEP-NECs. This study also highlights how a multi-parametric tool can capture clinically relevant prognostic subgroups better within the EP-NEC family, as compared to a single-variable assessment. In the present study, given the retrospective nature of the data collection in a multi-institutional setting over a number of years, collection of biochemical markers included in the GI-NEC score such as ALP and LDH was not possible, nor checked routinely in included patients; whether the incorporation of the morphological subtype in the GI-NEC score could further improve the prognostic performance of the model should be explored in a prospective setting.

Currently, the expression of NE markers and Ki-67 index are mandatory requirements for the formulation of an EP-NEC diagnosis [1], whereas the morphological subtype is reported at the discretion of the pathologist. One reason for this is that discrimination between a SC and a non-SC morphology in EP-NECs is often challenging, due to the frequent co-presence of features which are typical of both entities, especially when the amount or quality of tumour tissue is limited. Nevertheless, here, we propose that, when possible, the morphological subtype of EP-NEC should be determined, as it may yield critical information for the patient management.

The present study included data on one of the largest EP-NEC series from the current literature with available information on the morphological subtype. The retrospective nature of the data collection, holding possible inter-centre inconsistencies in the recording of clinical data, and the lack of central pathological review represent the two major limitations of this study. It is important to highlight the expertise in the field of the pathologists involved in this study as well as the application of uniform histopathological criteria in adherence with the most updated classifications of NENs. In addition, equivocal cases (identified at discretion of the individual pathologist reviewing the sample) were discussed among pathologists, and those on which an agreement was not achieved were discarded to minimise selection biases. While acknowledging these limitations, this study suggests that the morphological subtype is an independent prognosticator of PFS and OS in patients with advanced-stage EP-NEC and a predictor of response to first-line platinum/etoposide chemotherapy. In particular, patients with advanced-stage non-SC EP-NEC seem to have a worse prognosis than patients with advanced-stage SC EP-NEC, and this may be explained by the fact that the former subgroup has a lower chance of achieving disease control with first-line palliative treatment (with platinum/etoposide being the most commonly administered).

Similar to what has been found in other retrospective series [7,9,10,11,12], in the present study, a Ki-67 index of <55% was associated with a longer PFS and OS in the unadjusted model, although it did not affect response to first-line palliative treatment. Despite having a higher proportion of cases with a Ki-67 index of <55% compared to for the SC subgroup, the non-SC subgroup was associated with worse survival outcomes in the advanced-stage setting in the adjusted model; this is because the non-SC subgroup was nevertheless composed of a majority of cases with a Ki-67 index of ≥55%, which ultimately drove the prognosis of the whole subgroup.

Therefore, we wanted to explore how the combination of morphological subtype and Ki-67 index (<55% vs. ≥55%) determined survival outcomes of patients with advanced-stage EP-NEC. After demonstrating that these two variables impact in an independent way (interaction test: non-significant) on both PFS and OS, we stratified the advanced EP-NEC cohort in four subgroups. Subgroup (A); non-SC EP-NECs with a Ki-67 index of <55% had the most favourable prognosis. This might be a reflection of their intrinsic and more indolent biology and/or reduced platinum-sensitivity, and alternative first-line treatment strategies to platinum/etoposide chemotherapy may further improve the outcomes of this subgroup. It can be argued that this subgroup might include some G3-WD-NETs. In fact, similar to G3-WD-NETs [27], the majority of non-SC EP-NECs with a Ki-67 index of <55% from the present study were of pancreatic origin (52.2%) (Other sites of origin were stomach (13.0%), CUP (8.7%), colon (8.7%), small bowel (8.7%), ovary (4.3%) and biliary tract (4.3%)). Despite the exclusion of equivocal cases (NEC vs. G3-WD-NET) by pathologists, there is still the possibility that some tumours with a low proliferative rate (20% < Ki-67 < 50–55%) were misclassified as NECs rather than G3-WD-NETs, especially those with a non-SC morphology. In fact, the distinction between these entities can be challenging, especially in the absence of molecular data supporting one or the other diagnosis; mutation in *TP53* and/or *RB1* favours a NEC, whereas the mutation of the death-domain-associated protein (*DAXX*), alpha-thalassemia/mental retardation, X-linked (*ATRX*) or multiple endocrine neoplasia type 1 (*MEN1*) favours a G3-WD-NET diagnosis [28]. This issue is recognised within the clinical and scientific community with an interest in NENs [29]. Subgroup (B); non-SC EP-NECs with a Ki-67 index of ≥55% yielded the worst prognosis within the whole EP-NEC family; a possible explanation for the poor outcomes of this subgroup is its reduced platinum-sensitivity and alternative non-platinum/etoposide-based chemotherapy regimens or different treatment strategies, especially inclusion in clinical trials, should be considered for this subgroup. For example, the FOLFIRINEC study will compare platinum/etoposide vs. modified FOLFIRINOX as first-line chemotherapy for metastatic GEP-NEC, associated with molecular profiling and predictive biomarker identification (https://clinicaltrials.gov/ct2/show/NCT04325425, accessed on 30 June 2021). In addition, the data from the present study, although non-statistically significant and mostly derived in the second-line setting, indicates that a fluoropyrimidine/irinotecan combination could be more effective in non-SC EP-NECs. This is being explored in two randomised phase II studies: BEVANEC is evaluating the FOLFIRI regimen +/− bevacizumab as a second-line treatment after the failure of platinum-etoposide regimen in 124 patients with advanced GEP-NEC [30]; the NET-02 study is investigating a 5-fluorouracil/folinic acid/liposomal irinotecan combination or docetaxel as a single agent in 102 patients with available EP-NEC tissue who have failed platinum-based chemotherapy (at https://clinicaltrials.gov/ct2/show/NCT03837977, accessed on 30 June 2021). The NET-02 trial will provide insights on the activity of these two alternative regimens in EP-NECs, and future post-hoc analyses may unveil associations with the morphological subtype [31]. While little can be concluded about subgroup (C); SC EP-NECs with a Ki-67 index of <55% due to being numerically very small, subgroup (D); SC EP-NECs with a Ki-67 index of ≥55% included patients with a high chance of benefitting from platinum/etoposide and for whom this regimen still represents a valid option.

Although advanced EP-NEC subgroups according to combined morphological subtype and Ki-67 index (<55% vs. ≥55%) have shown prognostic significance, even after adjustment for the other independent prognosticator of PFS and OS (ECOG PS), the corresponding Kaplan–Meier curves did not show robust separation. This might be explained by the small number of subgroup (C); SC EP-NECs with a Ki-67 index of <55% (*n =* 5) alongside other potential biases related to the retrospective nature of the study, the lack of central pathological review and the possibility of some misdiagnoses, as discussed before. The interpretation of the stratification proposed in this study is, at the current stage, only “hypothesis generating” and warrants validation in a prospective setting, possibly a clinical trial, where those potential confounding factors can be better controlled and an adequate power calculation can allow reliable pre-planned subgroup analyses.

To conclude, this study provides a signal of the prognostic impact of the morphological subtype in patients with EP-NEC, especially within the subgroup with a Ki-67 index of ≥55%, where the Kaplan–Meier curves for both PFS and OS diverge more neatly; it suggests that reporting the morphological subtype should be considered as in the diagnostic work-up of patients with EP-NEC, as it provides complementary prognostic information to the Ki-67 index, and can potentially orientate the decision towards treating a patient in the first-line palliative setting with standard-of-care platinum/etoposide chemotherapy or rather attempting alternative treatment strategies. Next-generation sequencing studies have unveiled novel potential therapeutic targets for patients with EP-NECs, such as frequent *BRAF^V600^* mutation in colorectal NECs [32,33], microsatellite instability in GEP-NECs [12], and *MYCN*/*Aurora Kinase A* amplification in prostate NECs [34], paving the way for the use of targeted therapies and immunotherapy in this population. Ongoing efforts to continue characterising the EP-NEC molecular and immune landscape are expected to clarify whether critical biological diversities underpin the two morphological subtypes, supporting the need, or not, for addressing these two entities separately at a clinical level. In addition, there is known heterogeneity across EP-NEC sites of origin with regard to survival outcomes, proportion of SC and non-SC subtypes [2] and genomic features [28]. In fact, EP-NECs at different sites of origin have a variable prevalence of “site-specific” molecular features, mainly shared with non-neuroendocrine cancers from the same organs, as well as of *TP53* and *RB1* mutations (virtually ubiquitous in SCLC) [8,35,36,37]. It would be interesting to explore whether and how the morphological subtype plays a role in generating this inter-site biological diversity.

## Figures and Tables

**Figure 1 cancers-13-04152-f001:**
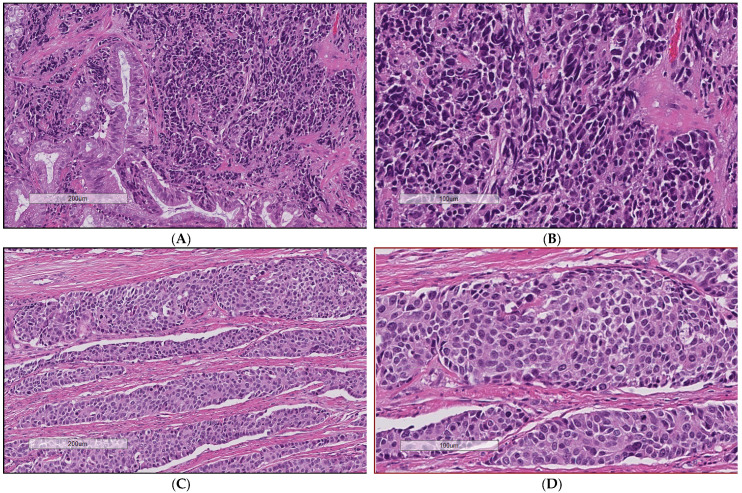
Examples of haematoxylin and eosin staining of extra-pulmonary neuroendocrine carcinomas. (**A**,**B**) Small-cell neuroendocrine carcinomas of the pancreas; sheets of packed tumour cells with fusiform nuclei and scant cytoplasm. (**C**,**D**) Large-cell neuroendocrine carcinomas of the colon; nests of polygonal tumour cells with prominent nucleoli and more abundant cytoplasm compared to their small-cell counterpart.

**Figure 2 cancers-13-04152-f002:**
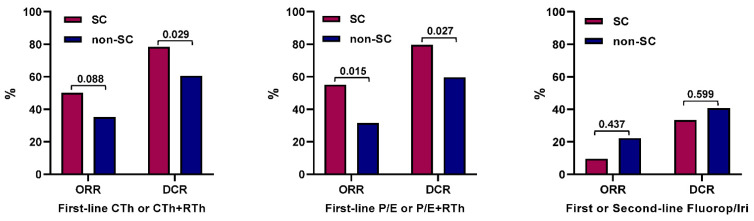
Response to treatment according to the morphological subtype. ORR, objective response rate; DCR, disease control rate; CTh, chemotherapy; RTh, radiotherapy; P/E, platinum/etoposide; Fluorop/Iri, fluoropyrimidine/irinotecan. *P*-values determined by Pearson’s chi-squared test or Fisher’s exact test are shown.

**Figure 3 cancers-13-04152-f003:**
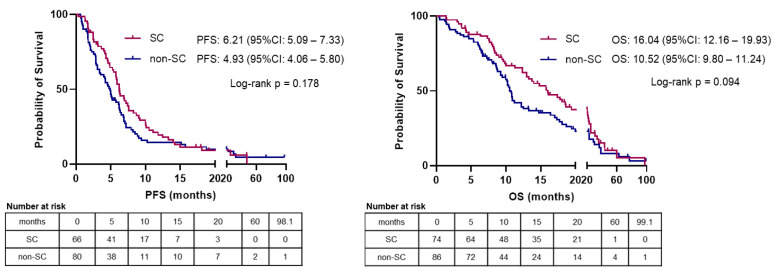
Kaplan–Meier curves for progression-free survival (PFS) and overall survival (OS) according to the morphological subtype in patients with advanced-stage extra-pulmonary neuroendocrine carcinoma in this study. PFS and OS in the graphs refer to median values calculated by Kaplan–Meier analysis. Log-rank *p* = *p* value calculated by applying Log-rank test (Mantel-Cox) for the equality of survival distributions.

**Figure 4 cancers-13-04152-f004:**
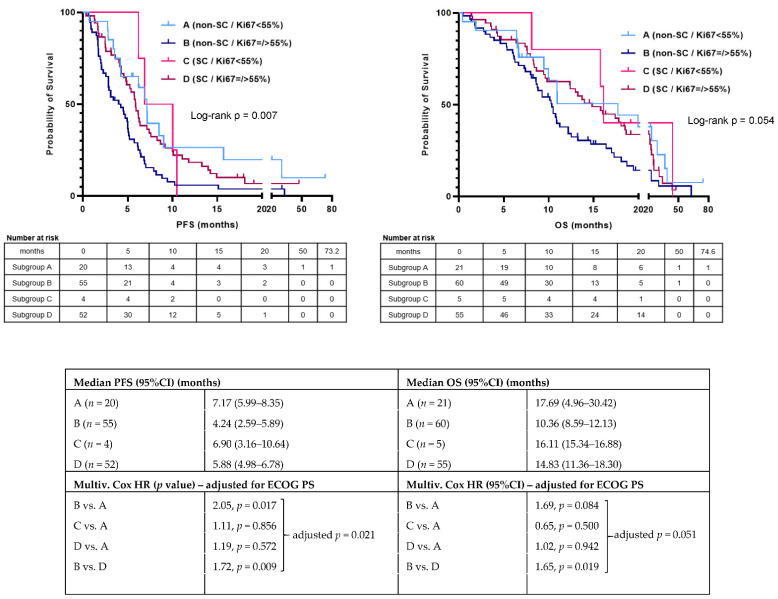
Kaplan–Meier analysis for PFS and OS according to subgroups per morphological subtype and Ki-67 index (<55% vs. ≥55%) in patients with advanced-stage extra-pulmonary neuroendocrine carcinoma in this study. *n* = number of patients; 95% CI, 95% confidence interval; ECOG PS, Eastern Cooperative Oncology Group performance status.

**Table 1 cancers-13-04152-t001:** Clinico-pathological characteristics of patients with extra-pulmonary neuroendocrine carcinomas included in this study.

		Whole Population (*n* = 170)	Small Cell (*n* = 77)	Non-Small Cell (*n* = 93)	*p*-Value
Characteristics		*n* (Frequency)	*n* (Frequency)	*n* (Frequency)	
**Gender**					
	Male	112 (65.9%)	53 (68.8%)	59 (63.4%)	*p* = 0.461
	Female	58 (34.1%)	24 (31.2%)	34 (36.5%)	
**Age at an initial diagnosis**					
	Median (range) (yrs)	62.5 (26.1–90.9)	58.8 (32.7–82.5)	64.8 (26.1–90.9)	
	Mean (StDev)	61.1 (12.7)	59.4 (11.5)	62.3 (13.4)	*p* = 0.140
**Site of origin**					
	Pancreas	41 (24.1%)	19 (24.7%)	22 (23.7%)	Oesophagus/OGJ vs. others (*p* = 0.001) Rectum vs. others (*p* = 0.068)
	CUP	39 (22.9%)	14 (18.2%)	25 (26.9%)	
	Colon	21 (12.4%)	7 (9.1%)	14 (15.1%)	
	Stomach	20 (11.8%)	6 (7.8%)	14 (15.1%)	
	Oesophagus/OGJ	12 (7.3%)	11 (14.3%)	1 (1.1%)	
	Rectum	11 (6.5%)	8 (10.4%)	3 (3.2%)	
	Biliary tract	11 (6.5%)	6 (7.8%)	5 (5.4%)	
	Small bowel	5 (2.9%)	2 (2.6%)	3 (3.2%)	
	Anus	3 (1.8%)	1 (1.3%)	2 (2.2%)	
	Bladder	2 (1.2%)	1 (1.3%)	1 (1.1%)	
	Head & Neck	2 (1.2%)	1 (1.3%)	1 (1.1%)	
	Prostate	1 (0.6%)	1 (1.3%)	0 (0.0%)	
	Ovary	1 (0.6%)	0 (0.0%)	1 (1.1%)	
	Appendix	1 (0.6%)	0 (0.0%)	1 (1.1%)	
	GEP	125 (73.5%)	60 (77.9%)	65 (69.9%)	GEP vs. CUP + others (*p* = 0.238) CUP vs. GEP + others (*p* = 0.179)
	CUP	39 (22.9%)	14 (18.2%)	25 (26.9%)	
	Others	6 (3.5%)	3 (3.9%)	3 (3.2%)	
**Disease stage at diagnosis ^#^**					
	Stage II	6 (3.5%)	2 (2.6%)	4 (4.3%)	
	Stage III	27 (15.9%)	8 (10.4%)	19 (20.4%)	
	Stage IV	131 (77.1%)	64 (83.1%)	67 (72.0%)	
	Potentially curable n.o.s.	6 (3.5%)	3 (3.9%)	3 (3.2%)	
	Potentially curable	39 (22.9%)	13 (16.9%)	26 (28.0%)	*p* = 0.087
	Incurable	131 (77.1%)	64 (83.1%)	67 (72.0%)	
**Ki-67 index**					
	Median (range) expressed in %	80.0 (25–100)	80.0 (25–100)	77.5 (25–100)	
	Mean (StDev)expressed in %	73.2 (19.9)	78.7 (18.5)	69.3 (20.1)	*p* = 0.002
	<55%	30/150 * (20%)	7/62 * (11.3%)	23/88 * (26.1%)	*p* = 0.025
	≥55%	120/150 * (80%)	55/62 * (88.7%)	65/88 * (73.9%)	
	Unknown	20 (11.8%)			
**Smoking history**					
	Active/former smoker	72/115 * (62.6%)	31/54 * (57.4%)	41/61 * (67.2%)	*p* = 0.278
	Never smoker	43/115 * (37.4%)	23/54 * (42.6%)	20/61 * (32.8%)	
	Unknown	55 (23.4%)			

*n*, number of patients; GEP, gastro-entero-pancreatic tract; OGJ, oesophagogastric junction; CUP, cancer; StDev, standard deviation; potentially curable, managed with curative intent; incurable, managed with palliative intent; potentially curable n.o.s. (not otherwise specified), managed with curative intent, however, information available does not allow for allocation to a precise disease stage. ^#^ American Joint Committee on Cancer (AJCC) Tumor Node Metastasis (TNM) cancer staging, 8th edition (2017). * number of patients for whom the information was available. *p*-value was determined by Pearson’s chi-squared test, Fisher’s exact test, unpaired T test or Mann–Whitney U test, as appropriate.

**Table 2 cancers-13-04152-t002:** Univariable and multivariable Cox-regression analysis for PFS and OS in patients with advanced extra-pulmonary neuroendocrine carcinoma in this study.

		PFS		OS	
		Univariable	Multivariable	Univariable	Multivariable
		HR, *p*-Value	HR, *p*-Value	HR, *p*-Value	HR, *p*-Value
		*n =* 146		*n =* 160	
Age at diagnosis ^#^	Continuous	1.00, *p* = 0.566		1.00, *p* = 0.838	
Gender	Male vs. Female	0.83, *p* = 0.317		0.88, *p* = 0.485	
ECOG PS	≥2 vs. 0–1	2.62, *p* < 0.005	1.99, *p* = 0.006	2.83, *p* < 0.005	2.36, *p* = 0.001
Ki-67 index	≥55% vs. <55%	1.67, *p* = 0.040	1.80, *p* = 0.028	1.54, *p* = 0.091	1.67, *p* = 0.063
Site of origin	CUP vs. GEP	1.01, *p* = 0.959		1.21, *p* = 0.576	
	Others vs. GEP	0.89, *p* = 0.821		0.92, *p* = 0.851	
Morphological subtype	non-SC vs. SC	1.27, *p* = 0.180	1.62, *p* = 0.016	1.35, *p* = 0.095	1.64, *p* = 0.015
Interaction Ki-67/morphological subtype		0.54, *p* = 0.300 *		0.79, *p* = 0.699 *	

*n* = number of patients; HR, hazard ratio; 95% CI, 95% confidence interval; ECOG PS, Eastern Cooperative Oncology Group performance status; GEP, gastro-entero-pancreatic tract; CUP, cancer of unknown primary. ^#^ at diagnosis of advanced stage disease. * interaction test calculated by Cox-regression analysis including also the two individual variables.

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
