# Peer review of "Is the Morphological Subtype of Extra-Pulmonary Neuroendocrine Carcinoma Clinically Relevant?"

_cancers, 2021, doi:10.3390/cancers13164152_

Round 1

Reviewer 1 Report

In this reviewer opinion, Authors response do not satisfy criticisms that have been previously highlighted.

Reviewer 2 Report

I am satisfied with the answers the authors provided and how they modified the manuscript. I have no further comment.

This manuscript is a resubmission of an earlier submission. The following is a list of the peer review reports and author responses from that submission.

Round 1

Reviewer 1 Report

The authors describe a four-institute collection of non-pulmonary small cell or NE tumors. This is a large cohort, although the heterogeneity reduces the ability to deduce significant conclusions.

The methods mention response assessment by recist. If this is based on retrospective chart review, the value of this assessment is not high. An emphasis is given to the analysis of disease control rate, but no explanation is given for the definition of stable disease (minimal interval required for this definision).

The statistical analysis identifies the histologic subtype (SC or non-SC) as significantly  associated with OS and PFS , both in univaritate and in multivariate analyses. Ki67 was marginally associated with benefit, thus the authors present a score combining KI67 and histologic subtype. The survival curves of the 4 resulting subgroups do not demonstrate robust separation.

A technical and important note - KM curves need to include "number at risks" bellow the x-axis.

Overall, this is an interesting and potentially important cohort, but with some additional effort, can become more significant. Specifically, some simple clinical parameters such as those utilized for the GI-NEC score could be easily acquired and the analysis expanded to include them.

The detailed description of treatments given regarding each treatment line for the localized and extensive disease patients is interesting but not very important and can be moved to supplementary data.

Author Response

Please see file upload entitled "Responses to Reviewers' comments".

Reviewer 2 Report

Frizziero et al report of a large series of extra-pulmonary poorly-differentiated neuroendocrine carcinoma patients (EP-NEC) and of their outcome on several different treatment regimens. The Authors with their work surprisingly found that patients with non-SC NEC have worse prognosis than those with SC NEC. As this finding could be of paramount interest for clinical and research purposes, the work must be cleared of potential biases. 

First, it should be clarified how tumors with ki67 <55% were included as poorly differentiated as this is an unusual finding. This is especially crucial for the "C" group with low ki67 and SC morphology (N=4), but also in the group with low ki67 and non-SC morphology (N=20), which taken together comprise about 15% of the total cohort. Moreover, in both the SC and non-SC tumor group the lower bound of the ki67 range is 25, which is unusually low for a poorly-differentiated neuroendocrine carcinoma. All of this could be the result of either sampling errors (inhomogeneous highly necrotic tumors, with biopsy performed to the more external part of the tumor) or misdiagnosis. The methodology by which this tumors have been included should be clarified or excluded from the analysis if the diagnosis cannot be certain. To the same end, patients who received RT together with platinum-etoposide chemotherapy should be removed from the ORR/DCR, PFS and OS analysis.

Author might consider improving readability of tables and of figures, in particular of the Sankey diagrams

Table 1: chi-squared/Fisher tests should be corrected for multiple comparisons where appropriate (e.g. site of origin)

Overall data presentation should be improved as it is confusing as it is in the present form. 

Author Response

(The authors gave the same response as above.)

Reviewer 3 Report

For Authors:

Only minor corrections and suggestions for changes and clarifications in the original MS.

Page 2

Line 97 UNK should be rather be named CUP – Cancer of Unknown Primary

Page 6 table 1

Why Oesophagus/OGJ vs. others, authors used Pearson’s chi2 test and for Rectum vs. others

Used Fisher’s exact test?

General more comfortable to the readers will be just removed continue data like median and mean age at initial diagnosis and Ki-67 index in single panel and then others (different test applied in the analysis).

Page 8

Table 2

CT mean also computed tomography maybe authors used Ctx as chemotherapy and Rtx as radiotherapy, will be much more clear for general readers not only oncologists.

Pgae 12

Line 298 should be: sub-population

Page 12

Figure 4

PFS or OS is always presented as median value, so no make sense described as mPFS or mOS on the graphs.

The same in others figures with K-M curves in supplementary Figure 1s and figure 2s.

Page 13.

The table 4 should be editorially corrected.

Page 15

Line 395 Can the authors expand the list of non-SC/Ki-67<55% origins to the other predominant locations, as they indicated pancreatic predominance 52.4%, and the remaining cases?

Page 17

References should be corrected as required by the editors of Cancers

  1. Busico, A., et al.,
  2. Gerard, L., et al.,
  3. Lamarca, A., et al.,
  4. Liu, A.J., et al.,
  5. Capdevila, J.,

Author Response

(The authors gave the same response as above.)

Reviewer 4 Report

Frizziero M et al. performed a multicentre retrospective study investigating potential differences in clinico-pathological characteristics and treatment/survival outcomes between small cell (SC) versus non-SC extra-pulmonary neuroendocrine carcinomas (EP- NECs).  Based on the analysis of 170 patients with EP-NEC, they suggested that the morphological subtype of EP-NEC provides complementary information to the Ki-67 index and aids identification of patients who could benefit from alternative first-line treatment strategies to platinum/etoposide chemotherapy.

The content is interesting and would be one answer to the question whether the morphological subtype (SC vs. non-SC) is relevant for management of EP-NEC patients.  However, there are some weaknesses as shown below.

  1. Figures 3 and 4 are the results of Kaplan-Meier analysis. The Kaplan-Meier analysis is a univariate approach, therefore, it is not adequate to add the results of multivariate analysis and the analysis should not be conducted using Cox-regression model.  Moreover, which test was used is not described in the Materials and Methods section. Log-rank test? Wilcoxon test?

  1. Figure 1 lacks scale bars. Please add them.  Moreover, photos showing a higher magnification should be also added because it is hard to recognize nucleoli.

  1. Both “p-value” and “p value” are observed in the manuscript. Please use the same terminology throughout the manuscript.

  1. In line 347, 31.58% should be revised to 31.6%. In line 353, 9.52% should be revised to 9.5%.  Please check the others throughout the manuscript.

  1. In lines 58 and 59 “worse progression-free survival (PFS) (adjusted-HR=1.615, p=0.016) and overall survival (OS) (adjusted-HR=1.640, p=0.015)”, 1.615 and 1.640 are not described elsewhere. I think that 1.615 and 1.640 are calculated by 1/0.62 and 1/0.61, respectively, based on the Table 2. Please add these numbers in Result section if you use them in abstract.

  1. In line 586, itsuggests -> it suggests

Author Response

See file attached with point-by-point replies to teh reviewer's comments

Reviewer 5 Report

In this manuscript by Frizziero et al., the authors looked in a multi center cohort at the differences between non-lung SC and NSC NEC and their impact.

While this study has the merit of the number and the efforts that a multicenter study requires I have some points that I feel needs to be addressed or discusses:

-  in the advanced+ potentially curable  groups the number is >170. Are there patients that are in both groups?

-Regarding the Path review. Zoom meetings are mentioned and case exclusion if an agreement could not be reached. How many cases were excluded? Please cite the pathology criteria used for SC and NSC, especially regarding the neuroendocrine markers.

The lack of central path review is a major limitation and needs to be discussed as such.

-Remove the KM curves with the group <55%, as there is not enough patient to draw any conclusion and possibly well differentiated agressive NET.

- The authors should discuss more than NSC NEC are different biologically depending on the organ of origin (SCC seems more homogeneous). What is know about the biological differences between NSC NEC from pancreas colon etc...

- Did the authors tested in univariate analysis the impact of the organ of origin?

- The conclusions may need to be a bit "waterdown"

Author Response

See file attached with point-by-piont replies to the reviewer's comments.

Round 2

Reviewer 1 Report

Most of the comments have been adequately answered. 

Author Response

See file attached with poit-by-point replies to the reviewer's comments.

Reviewer 2 Report

Authors did not follow suggestion or provide acceptable justification for manuscript analysis that is therefore unsuitable for publication in this reviewer opinion.

Author Response

See file attached with point-by-point replies to the reviewer's comments.
